# *Mycobacterium abscessus*: It’s Complex

**DOI:** 10.3390/microorganisms10071454

**Published:** 2022-07-19

**Authors:** Hazem F. M. Abdelaal, Edward D. Chan, Lisa Young, Susan L. Baldwin, Rhea N. Coler

**Affiliations:** 1Center for Global Infectious Disease Research, Seattle Children’s Research Institute, Seattle, WA 98145, USA; hazem.abdelaal@seattlechildrens.org (H.F.M.A.); susan.baldwin@seattlechildrens.org (S.L.B.); 2Department of Academic Affairs and Medicine, National Jewish Health, Denver, CO 80206, USA; chane@njhealth.org; 3Pulmonary Section, Rocky Mountain Regional Veterans Affairs Medical Center, Aurora, CO 80045, USA; 4Department of Medicine, Johns Hopkins School of Medicine, Baltimore, MD 21205, USA; lyoung1090@gmail.com; 5Department of Pediatrics, University of Washington School of Medicine, Seattle, WA 98195, USA; 6Department of Global Health, University of Washington, Seattle, WA 98195, USA

**Keywords:** *Mycobacterium abscessus*, pulmonary infection, host–pathogen interaction, novel therapeutics, pathogenesis

## Abstract

*Mycobacterium abscessus* (*M. abscessus*) is an opportunistic pathogen usually colonizing abnormal lung airways and is often seen in patients with cystic fibrosis. Currently, there is no vaccine available for *M. abscessus* in clinical development. The treatment of *M. abscessus*-related pulmonary diseases is peculiar due to intrinsic resistance to several commonly used antibiotics. The development of either prophylactic or therapeutic interventions for *M. abscessus* pulmonary infections is hindered by the absence of an adequate experimental animal model. In this review, we outline the critical elements related to *M. abscessus* virulence mechanisms, host–pathogen interactions, and treatment challenges associated with *M. abscessus* pulmonary infections. The challenges of effectively combating this pathogen include developing appropriate preclinical animal models of infection, developing proper diagnostics, and designing novel strategies for treating drug-resistant *M. abscessus*.

## 1. Introduction

Non-tuberculous mycobacteria (NTM) are environmental bacteria that are found ubiquitously in the soil and in natural and potable water, and biofilms. While human exposure to NTM is widespread, diseases caused by them are relatively uncommon because of the low pathogenicity of most NTM. However, due to NTM prevalence in natural and human-made environmental niches in combination with host risk factors, NTM infections are becoming an emerging global health concern in several countries. Clinical manifestations range from cutaneous infections to chronic lung disease to disseminated disease. Of the approximately 200 species of NTM identified, most infections are caused by *Mycobacterium avium* (*M. avium*) complex (MAC, comprised of several species, with the most common being *M. avium*, *M. intracellulare*, and *M. chimaera*, among several others), *M. abscessus* complex (with subspecies *abscessus*, subsp. *massiliense*, and subsp. *bolletii*), and *M. kansasii* [1]. However, depending on the region of the world, some other “less common” NTM species may be relatively prevalent such as *M. malmoense* in Northern Europe and *M. xenopi* in Canada and certain regions of Europe (central, southern, and the United Kingdom). Lung disease due to NTM is estimated to be increasing by approximately 8% per year in the United States in 2.3 million Medicare beneficiaries that were over 65 years of age, and in whom 58% were female subjects [2]. Other regions of the world have also noted an increase prevalence of NTM lung disease (NTM-LD). For example, NTM cases also increased from 0.9 to 2.9 per 100,000, respectively, from 1995 to 2006 in England, Wales, and Northern Ireland [3] and are also increasing in Canada [4]. The two main categories of NTM include the rapidly growing mycobacteria and the slowly growing mycobacteria. The rapid growers include the *Mycobacterium chelonae-abscessus* complex, including *M. abscessus* subsp. *abscessus*, *M. abscessus* subsp. *bolletti*, *M. abscessus* subsp. *massiliense*, *M. chelonae*, and *M. fortuitum* [5]. The opportunistic slow growers include the *M. avium* complex (MAC), with several different species including *M. avium*, *M. intracellulare* subsp. *intracellulare*, and subsp. *chimaera*, among others.

As an opportunistic pathogen, *M. abscessus* often colonizes the lung airways in patients with cystic fibrosis (CF), chronic obstructive pulmonary disease (COPD), or bronchiectasis [6], whereas humans (and mice) with normal lung airways are typically resistant to lung infection. Much more remains to be discovered about the host–pathogen interactions associated with a progressive pulmonary parenchymal infection.

The intrinsic drug resistance in *M. abscessus* has limited its therapeutic options, with only a few active antibiotics available that are effective. Severe *M. abscessus* lung infections are often treated with a cocktail of oral and parenteral antibiotics as well as surgical lung resection in those with localized but recalcitrant lung disease.

An evaluation of the in vivo susceptibility and efficacy of these drugs against acute *M. abscessus* infection has been performed using both immunocompromised mice and zebrafish embryos. Imipenem-cilastatin and clarithromycin conferred protection against *M. abscessus* as demonstrated by restricted mycobacterial growth, decreased pathologic manifestations such as brain abscesses, and increased embryo survival [7]. However, antibiotic use has been limited due to non-standardized in vivo antibiotic susceptibility testing, limited cellular/animal testing models, non-uniform susceptibility among different clinical isolates of *M. abscessus*, and incongruent clinical response, often resulting in long-term failure [8]. Worldwide, it is plausible that increased outbreaks and nosocomial transmission of *M. abscessus* complex are rising, reflecting an emerging antibiotic resistance crisis that threatens public health [9,10]. Further research is needed to identify optimal in vivo conditions to test the drug efficacy and optimize therapeutic regimens for *M. abscessus* infections.

## 2. Critical Elements of *M. abscessus* Virulence and Pathogenicity

Classifying *M. abscessus* as an opportunistic pathogen occurring in immunocompromised individuals has recently been challenged based on the observation that the *M. abscessus* clade, with numerous virulence factors, may actually fit the criteria of a true pathogen [11]. The ability of *M. abscessus* to thrive in various environmental niches [12] helps explain its ability to also colonize human-made sources and materials, such as potable water and plumbing systems, as well as medical and surgical instruments/devices resulting in nosocomial NTM infections [13,14,15].

The increasing incidence of *M. abscessus* infections in cystic fibrosis (CF) and non-CF populations reported globally [16] provides further evidence that *M. abscessus* possesses a diverse repertoire of virulence factors that are representative of a successful human pathogen [11]. *M. abscessus* pathogenic evolution may be forced by the competing requirement to maintain transmission fitness [17]. Perhaps the most prominent finding in these evolution-based studies are the results that identified strong evolutionary pressure on *M. abscessus* genes promoting survival within the macrophage, which seem to come with a fitness cost in favor of enhanced virulence of *M. abscessus* isolates. 

Due to the ability of *M. abscessus* to survive and replicate within free-living amoeba, an essential role of the ESX-4 *M. abscessus* type VII secretion system (T7SS) was discovered, further substantiating that a genetic factor may have naturally selected for intracellular survival of *M. abscessus* [18]. ESX-4 in *M. abscessus* is known to block phagosomal acidification and disrupt phagosomes, similar to the role of ESX-1 in *Mycobacterium tuberculosis* (*M. tuberculosis*) [18]. In humans and other animal species, the transcriptomic changes of *M. abscessus* during intracellular growth in macrophages have demonstrated upregulation of genes such as heat shock and oxidative stress (e.g., GroEL-ES and hsp) to cope with intracellular stresses [19]. Those factors are associated with a switch to a slower growth phenotype, and the utilization of fatty acids as an energy source, which displays the range of mechanisms *M. abscessus* uses to persist in the host. An additional important virulence factor includes the complex lipids of *M. abscessus*, including glycopeptidolipids (GPLs), involved in adherence of *M. abscessus*. The loss of GPL is associated with the transition from a smooth (S) to a rough (R) morphotype in NTM [20,21], resulting in increased virulence in part by forming large extracellular serpentine cording, leading to the prevention of phagocytosis, increased inflammation, and the characteristic formation of abscesses [22,23]. Interactions between *M. abscessus* and host myeloid cells revealed the utilization of this extracellular cording to subvert the host innate immune response [7]. Cord-deficient *M. abscessus* mutants exhibit altered mycolic acid composition as well as extremely diminished phenotypes (i.e., lack of granuloma formation and replication in macrophages) in wild-type and immunocompromised zebrafish embryos [24]. GPLs also inhibits apoptosis in *M. abscessus*-infected macrophages by interacting with the mitochondrial cyclophilin D [25]. During persistent infection, the transition to this unique R colony morphotype is critical for invading and destroying a variety of cell types, including macrophages in monolayer culture, as seen in the zebrafish embryo infection model [5]. However, the cellular triggers for this transition between morphotypes remain unclear. 

Another role for GPL is triggering a strong humoral immune response [23]. Anti-GPL response may explain the conversion to the more virulent R morphotype in order to escape such response [23,26]. 

The involvement of complex lipids of *M. abscessus* in virulence has also shed light on the importance and role of lipid-specific transporters. For example, the Mycobacterial membrane protein Large (MmpL) permeases family, including MmpL4a and MmpL4b, are involved in GPL transport [27]. Another MmpL discovered to be associated with *M. abscessus* virulence is MmpL8 [28]. The loss of function (LOS) of MmpL8 results in reduced glycosyl diacylated nonadecyl diol (GDND) production and decreased intracellular survival and virulence. 

The growing incidence of *M. abscessus* can also be linked to the global rise in individuals who are immunocompromised or have pre-existing lung conditions [29]. *M. abscessus* can also cause extrapulmonary infections in the skin, soft tissue, bones, joints, lymph nodes, and internal organs [30]. Like its fellow pathogenic mycobacteria, *M. abscessus* can form distinctive granulomas by evading phagosomal defense mechanisms (macrophages and neutrophils), inducing the production of inflammatory cytokines, such as tumor necrosis factor (TNF), and recruiting B and T lymphocytes to the site of infection.

## 3. NTM Diseases

### 3.1. NTM Lung Disease in Subjects with Known Risk Factors

Non-tuberculous mycobacterial lung disease (NTM-LD) occurs primarily in three broad groups of patients (Figure 1): (i) Acquired: Those with acquired anatomic lung or immune abnormalities with no identifiable genetic basis; e.g., localized bronchiectasis from prior unrelated infections, smoking-related emphysema, pneumoconiosis such as silicosis [31,32,33] as well as the use of either inhaled glucocorticoids or tumor necrosis factor (TNF) antagonists [34], (ii) Genetic/Hereditary: Those with genetic disorders that predispose to bronchiectasis and/or lung infections; e.g., cystic fibrosis (CF), primary ciliary dyskinesia, alpha-1-antitrypsin (AAT) deficiency, Williams–Campbell syndrome, Mounier-Kuhn syndrome, Sjogren’s syndrome, pulmonary alveolar proteinosis (PAP), and common variable immunodeficiency [35,36], and (iii) No Obvious Predisposing Factors: Those with no known prior lung or immunological abnormalities [37,38,39]. The pathophysiologic mechanisms by which primary immunologic and/or genetic disorders predispose to isolated NTM-LD are listed in Table 1. One of the acquired risk factors includes chronic aspiration. Chronic aspiration due to gastroesophageal reflux disease (GERD) has been reported to be more common in patients with lung disease associated with MAC and other NTM [40,41,42]. MAC disease can also occur in patients that aspirate due to other reasons. Swallowing disorders can also lead to the risk of aspiration (Figure 1) [40].

Another acquired risk factor listed in Figure 1 is calcified chest adenopathy. In a retrospective study including 79 patients with pulmonary MAC disease, calcified chest adenopathy was present in one-third of the patients, and furthermore, those with calcified lymphadenopathy in the chest lived in a *Histoplasma capsulatum* endemic area, whereas those living outside this endemic region had no adenopathy [43]. This led to a hypothesis that fungal infection could predispose the lungs of some patients to MAC infection by either airway distortion or parenchymal damage [43]. 

Smoking is known to be a risk factor for TB, which is not surprising based on the damaging effects that smoking has on the lungs and the immune system (for a recent review, see Quan et al. [44]). Smoking-related emphysema is an acquired risk factor for NTM lung disease as well (Figure 1). PAP as a risk factor is supported by the finding that mice with disruption of the granulocyte-monocyte colony stimulating factor-1 are more vulnerable to *M. abscessus* [45]. We discuss below in greater detail two conditions well established to be associated with NTM-LD: CF and AAT deficiency.

#### 3.1.1. CFTR Anomalies and Susceptibility to NTM

CF is an autosomal recessive disorder caused by mutation of the CYSTIC FIBROSIS TRANSMEMBRANE CONDUCTANCE REGULATOR (*CFTR*) gene. The prevalence of a CFTR gene mutation in the Caucasian population is estimated to be 1 in 20 individuals, resulting in occurrences of CF in 1 in 2000–2500 live births [52]. CF patients are particularly susceptible to recurrent and chronic bacterial and mycotic infections, including those due to *Staphylococcus aureus*, *Pseudomonas aeruginosa*, *Burkholderia cepacia* complex, and NTM, among others [53,54,55]. The mechanisms for susceptibility in CF patients are complicated by thickened mucus and inability to clear bacteria, secondary ciliary dysfunction, and reduced human beta-defensin-2 function and level [56]. Macrophage dysfunction due to the *CFTR* mutation has also been demonstrated and resulting in impaired phagocytosis and reduced efferocytosis (with reduced clearance of apoptotic neutrophils), and excessive production of inflammatory mediators directed towards microbial stimuli [34,57]. Although the topic of whether the M1 (“pro-inflammatory”) or the M2 (“anti-inflammatory”) macrophage phenotype predominates in the CF lung is controversial, there is evidence that CF macrophages are defective in switching between these two phenotypes [58].

Heterozygous carriers of a single *CFTR* mutation may also be more susceptible to NTM lung infection, particularly with respect to bronchiectasis [59,60]. It is not known, however, whether the one defective *CFTR* gene increases the susceptibility to NTM, bronchiectasis, or both. 

#### 3.1.2. AAT Anomalies and Susceptibility to NTM

AAT deficiency predisposes to NTM-LD by two main mechanisms. One is that AAT deficiency is a risk factor for both chronic obstructive pulmonary disease and bronchiectasis, well-known underlying lung conditions for subsequent NTM-LD [34,56,57,61].

Another mechanism is that AAT itself has host-defense properties against NTM through induction of autophagy in macrophages [62]. This finding is supported epidemiologically by studies showing that the presence of heterozygous AAT anomalies—which are not known to cause bronchiectasis on their own—were more common in patients with NTM-LD compared to the general population in the U.S. [63,64]. Thus, susceptibility of AAT-deficient individuals to NTM-LD may occur because of impaired innate immunity as well as alterations in lung architecture (bronchiectasis and COPD).

### 3.2. NTM-LD in Patients without a Known Underlying Cause

In individuals with NTM-LD without any known predisposing conditions, the bronchiectasis, bronchiolitis, and the sequelae of atelectasis and cavities are presumed to be caused by the NTM infection and chronic airway inflammation. It has been observed that a significant number of NTM-LD subjects without an identifiable predisposing factor possess a life-long slender body habitus (Figure 1) [38,46,65,66,67] and reduced visceral fat with the caveat that weight loss may also occur post-NTM infection [68]. Low body weight itself has been hypothesized as a risk factor for NTM-LD and tuberculosis [35,69,70,71]. Furthermore, low BMI (<18.5 kg/m^2^) is associated with a greater number of diseased lung segments and NTM-LD-specific mortality [69,72]. Interestingly, NTM-LD has been reported in younger women (ages 20–53 years old) diagnosed with anorexia nervosa [73,74,75,76]. In general, since NTM-LD is much less common in younger individuals than in the elderly, this lends credence to the possibility that a thin body habitus is a risk factor for NTM-LD. Elderly Caucasian, post-menopausal females with slender body morphotypes are disproportionately predisposed to NTM-LD compared to males [47,54]. Hormonal changes and immune-senescence (changes in the immune system as a result of aging) likely contribute to risk factors for NTM-LD, in which an accumulation of damaged DNA and other factors are thought to result in a low-grade inflammatory phenotype termed ‘inflammaging’, which can alter immune function [47,49]. 

A possible mechanism by which slender individuals with low body fat content may be predisposed to NTM infections is the relative deficiency of leptin, a satiety hormone [77]. Leptin has a number of immunomodulatory functions that can potentially enhance host immunity against NTM, including the differentiation of uncommitted T0 cells toward the TH1 interferon-gamma (IFNγ)-producing phenotype [77]. Indeed, mice deficient in leptin are more susceptible to experimental *M. abscessus* lung infection [65,77,78]. Reduced levels of leptin in the sera of pulmonary NTM patients have also been observed [79].

Some NTM-LD patients have a greater than expected preponderance of abnormalities within the thoracic cage region, such as pectus excavatum and scoliosis [46,65,66,67,75,79,80]. We and others have postulated that thoracic cage abnormalities may be a marker for an underlying and yet-to-be identified genetic predisposition, perhaps related to a minor variant of Marfan syndrome (due to mutations of fibrillin-1) or ciliary dysfunction (due to mutations of different genes that encode for ciliary proteins) [65,66], [75,80,81,82]. Pectus excavatum and scoliosis have also been described in other connective tissue disorders, such as Loeys–Dietz syndrome (LDS, due to gain-of-function mutation of transforming growth factor-beta receptors 1/2—TGFβR1/2) and Shprintzen–Goldberg Syndrome (SGS, due to mutation of the Sloan Kettering Institute (SKI) protein, a downstream inhibitor of TGFβ signaling) [83]. While these disorders are due to monogenic mutations of different genes, each result in increased signaling of TGFβ, a cytokine known to predispose to NTM infection [84,85]. 

In light of this, the whole blood of NTM patients was found to produce more TGFβ, and lower levels of IFNγ upon ex vivo stimulation with various Toll-like receptor agonists or with *M. intracellulare* as compared to similarly stimulated whole blood from uninfected controls [65]. Daniels et al. analyzed for the presence of dural ectasia—an enlarged dural sac seen in MFS, LDS, and SGS—in patients with idiopathic bronchiectasis, CF subjects, MFS, and controls and found that the L1–L5 dural sac diameter was significantly greater in patients with idiopathic bronchiectasis as compared to controls and to CF subjects, suggesting the possibility of an underlying connective tissue disorder in those with idiopathic bronchiectasis [82]. They also found a strong correlation between dural sac size and NTM-LD, as well as dural sac size and long fingers [82]. NTM-LD was also reported in a patient with congenital contractural arachnodactyly, a genetic disorder due to FIBRILLIN-2 gene mutation and which shares many clinical features with MFS [86].

Fowler et al. described reduced ciliary beat frequency in the nasal epithelium and reduced nasal nitric oxide (NO) in NTM-LD patients compared to controls; the ciliary beat frequency was increased by NO donors or compounds that increased the concentration of cyclic guanosine monophosphate, a downstream mediator of NO [87]. Subsequent whole exome sequencing of NTM-LD subjects showed, compared to control data from the 1000G Project, increased variants in immune, CFTR, ciliary, and/or connective tissue genes, implicating a multigenic disorder for some patients with NTM-LD [81].

Because the variants of immune genes were significantly more common in NTM-LD patients than in unaffected family members, immune gene variants may be the discriminating genetic factor for the development of NTM-LD [81]. Furthermore, the number of CFTR variants per person was actually greater in both control groups (family members not infected with NTM and in the 1000G Project cohort) than in NTM-LD subjects [81]—in contrast to the other three non-CFTR gene categories in which the number of variants was least in the control 1000G Project cohort—it favors the possibility that the risk for NTM-LD in CF patients is perhaps due to the presence of bronchiectasis and not to the CFTR mutation per se. Becker and colleagues performed whole exome sequencing on 11 NTM-LD subjects with slender body habitus, pectus excavatum, and scoliosis and found one with mutation of the Fibrillin-1 gene and four (two being sisters) with heterozygous mutations of the Macrophage-Stimulating 1 Receptor (MST1R) gene and in none of 29 NTM-LD patients without pectus excavatum or scoliosis [88]. While these investigators showed that MST1R may function to increase IFNγ production, MST1R was previously reported to be a tyrosine kinase receptor found on the apical epithelial surfaces of fallopian tubes and airways and upon binding to its ligand (macrophage stimulating protein), enhanced ciliary beat frequency [89,90].

### 3.3. Disseminated NTM Disease 

Patients with extrapulmonary visceral organ or disseminated NTM disease are almost always frankly immunocompromised, such as those receiving tumor necrosis factor (TNF) antagonist therapy, organ transplantation, and having untreated AIDS (Figure 1 and Figure 2) [91,92,93]. Figure 2 includes several host-defense pathways used against *M. abscessus* in addition to mechanisms that can interfere with host-defense and lead to NTM disease. The use of immunosuppressive drugs, such as inhaled corticosteroids, can increase the risk of NTM disease [50,51] (Figure 1 and Figure 2). Mutations in GATA2 (guanine-adenine-thymine-adenine-2), a transcription factor, can lead to monocytopenia and mycobacterial (MAC) infection (called monoMAC syndrome), causing disseminated NTM with decreases in monocytes, DC’s, B cells and NK cells [94,95] (Figure 2). Individuals with other certain rare inherited disorders—particularly those with defects of the interleukin 12 (IL-12)/interferon-gamma (IFNγ) cytokine axis, and that fall under the rubric of Mendelian Susceptibility to Mycobacterial Diseases (MSMD)—are predisposed to an extrapulmonary visceral organ or disseminated NTM infections (Figure 2) [96,97,98,99,100,101,102,103,104,105,106,107,108]. Several of the MSMD-causing mutations have been identified in seven different autosomal genes and are described in detail in the review by Bustamante et al. [100]. Some of the proteins encoded by these genes are included in Figure 2 and represent the importance of each in the host’s defense against mycobacterial infections, including the IL-12 receptors (encoded by *IL12 B* (p40 subunit) and *IL12B1* (*b*1 chain of the IL-12 receptor), IFNγ receptors (encoded by *IFNGR1* and *IFNGR2*), transcriptional factor induced by IFNγ (*IRF8*), and signal transducer and activator of transcription 1 (*STAT1*); IFNγ-inducible factor (*ISG15*) (one of the seven autosomal genes with MSMD-causing mutations, not shown in Figure 2). MSMD-causing mutations in the X-linked gene include the nuclear factor-kappa B (NF-kB) essential modulator (*NEMO*), shown in Figure 2. The major component of the NADPH oxidase complex (*CYBB*) is another X-linked gene with MSMD-causing mutations [100]. Susceptibility to disseminated NTM in such individuals is corroborated experimentally by the increased vulnerability to *M. abscessus* in the IFNγ-knockout mice (Ordway et al., 2008). Individuals with acquired autoantibodies to IFNγ have more recently been described to be also more vulnerable to extrapulmonary visceral organ and disseminated NTM disease [109]. TNF-α inhibitors (including anti-TNF-α monoclonal antibodies and soluble TNF receptor fusion proteins (TNFR) used to suppress the immune response in patients with chronic inflammatory diseases, such as rheumatoid arthritis (RA), have been reported to increase the rate of mycobacterial disease, including NTM, compared to untreated patients and the general population [93,110,111]. The mechanism of anti-TNF monoclonal antibodies and soluble TNFR in host-defense impairment have been described elsewhere [112].

Many mycobacteria, including *M. abscessus* and fungi, are recognized by Toll-like receptor 2 (TLR2) and the beta-glucan receptor Dectin-1 [113]. Dectin-1 signaling leads to caspase-1 and IL-1β activation through the nucleotide-binding domain (NOD)-like receptor protein 3 (NLRP3)/ASC inflammasome, leading to host defense responses against *M. abscessus* [114]. Both Dectin-1 and TLR2 are necessary for *M. abscessus*-induced expression of innate antimicrobial responses, including interleukin-1 beta (IL-1β) and LL-37 [115]. TLR2-deficient mice are extremely susceptible to rough variants of *M. abscessus* due to failure of TH1-induced immunity [116]. Interleukin-8 (IL-8, also called CXCL8) is a chemokine-induced by infection and produced by macrophages and other cells and is a neutrophil chemotactic factor. Early neutrophil responses may help control infection with NTM, as shown with *M. fortuitum* [117]. These “experiments of nature” provide great insights into which elements of the immune system provide host-induced protection against mycobacteria. 

## 4. Treatment against *M. abscessus* Related Infections

### 4.1. Antibiotics Used for Treating M. abscessus

Despite its low virulence, treatment of *M. abscessus* is particularly difficult because of its intrinsic resistance to several commonly used antibiotics (Figure 3). Recommendations from The American Thoracic Society/Infectious Diseases Society of America include macrolides (typically Azithromycin favored over Clarithromycin) [118], Aminoglycosides (Amikacin), Carbapenems (Imipenem), and Cephamycins (Cefoxitin) [118,119,120].

Macrolides target bacterial 23S rRNA, inhibiting bacterial protein synthesis. *M. abscessus* possess two major forms of macrolide resistance, and both involve the bacterial 23S rRNA by different mechanisms. The first is “genetic” macrolide resistance and is due to single point mutation in position 2058 or 2059 of the bacterial 23S rRNA gene (also known as the *rrl* gene) [121]. The second form is known as “inducible” macrolide resistance, wherein a functional *ERM41* gene encodes a methylase that occupies a site on 23S rRNA preventing macrolides from binding [122,123] (Figure 4). Among the *M. abscessus* organisms, the majority of subsp. *abscessus* and subsp. *bolletii* strains possess a functional *ERM41* gene, which confers an inducible resistance to macrolides.

Conversely, a minority (15–20%) of subsp. *abscessus* isolates possess a T28C mutation of the *ERM41* gene, resulting in a non-functional methylase with preserved macrolide susceptibility. Similarly, all subsp. *massiliense* strains contain a partially deleted, non-functional *ERM41* gene and thus also have preserved macrolide susceptibility. Thus, in the absence of *rrl* gene mutation, NTM-LD patients infected with *M. abscessus* with a non-functional *ERM41* gene and hence preserved macrolide susceptibility (a minority of subsp. *abscessus* strains and all strains of subsp. *massiliense*) have better clinical outcomes than those infected with *M. abscessus* isolates with a functional *ERM41* gene and consequently inducible macrolide resistance (most subsp. *abscessus* and essentially all of subsp. *bolletii*).

The enzymatic modification of antibiotics by N-acetyltransferases confers aminoglycoside resistance. These specific enzymes add chemical groups to the 2′ amino groups of aminoglycosides, thus blocking the antibiotic from binding to its target protein [124]. In recent years, poor outcomes in patients infected with susceptible strains (approximately 90% of *M. abscessus* clinical isolates) stem from the development of resistance to amikacin, a key drug. In particular, patients with amikacin-resistant *M. abscessus*-LD (frequently involving *rrs* mutations) have shown unsatisfactory treatment outcomes, which is problematic given amikacin’s important role in long-term treatment [125]. *M. abscessus* can acquire fluoroquinolone resistance through cumulative mutations in a highly conserved region in the quinolone resistance-determining region (QRDR) of a DNA gyrase gene [126].

Unique structural and pathological traits contribute to drug resistance in clinical isolates of *M. abscessus*, including a capacity to form biofilms that prevents drug penetration [127]. A major pathogenic trait is an indolent progression; the rapid, silent growth can go undetected, eventually causing a severe deterioration in the human host [128]. Patients with chronic structural lung diseases such as CF and emphysema are at exceptionally high risk of pulmonary disease [129]. Early signs include sudden, progressive lung dysfunction, often accompanied by caseous lesions and alveolar granulomas [130,131]. While technically serving as a host defense mechanism, the granuloma also enables latent NTM infection and drug evasion by blocking drug penetration. Thus, early diagnosis and detection are vital but limited by poor understanding of its pathogenesis and the inability to adequately differentiate its symptoms from TB. Given the lack of standardized diagnostic criteria, misdiagnosis and treatment with anti-*Mycobacterium tuberculosis* medications are frequent but inappropriate considering the distinct treatment needs [126].

### 4.2. Strategies for Treating Drug-Resistant M. abscessus

Worldwide, outbreaks and nosocomial transmission of *M. abscessus* complex are rising, reflecting an emerging drug resistance crisis and a critical public health problem [9]. In recent years, poor treatment outcomes in patients infected with susceptible strains, which comprise approximately 90% of *M. abscessus* clinical isolates, have been spurred by the development of resistance to amikacin, a key drug against this pathogen. Given the current suboptimal outcome in patients with *M. abscessus* infection, more effective antimicrobials are needed not only for killing efficacy but also for a shorter time of treatment. 

Currently, macrolides (Clarithromycin or Azithromycin) are the most used antibiotics against *M. abscessus*. Thus, it is not surprising whether a *M. abscessus* isolate is susceptible or resistant to the macrolides is a key decision point in both the initial choice of antibiotic regimen and clinical outcome (Figure 3). While the absence or presence of macrolide resistance is also a key decision point for treatment and outcome of MAC-LD, the option for oral antibiotics for *M. abscessus* is much more limited than for MAC.

Several antibiotic alternatives to treat multi-drug resistant (MDR) *M. abscessus* include natural plant-derived products with antimicrobial effects, antimicrobial nanoparticles, antimicrobial peptides, antibiotic combinations, structurally modified antibiotics, pathogen-specific monoclonal antibodies, drug-induced changes in small regulatory RNAs (sRNAs), and therapeutic bacteriophages [132,133]. Of these, the use of pathogen-specific bacteriophages, known as phage therapy (PT), has shown exciting results. Abundant in nature and prolific, phages can either actively replicate (the lytic cycle) or lie dormant (the lysogenic cycle) in their hosts. Genetic engineering can be used to enhance the killing properties and host range of phages [132]. The first successful use of PT to treat a severe *M. abscessus* subsp. *massiliense* infection occurred in 2019 in a 15-year-old lung-transplant patient [134]. No adverse effects were observed following a cocktail regimen with three phages (one natural, two engineered). However, limitations to PT include a lack of lytic phages with an extensive host range and a possibility of emerging phage resistance. Bacteriophage cocktail therapy and CRISPR-Cas genomic technology are being strongly considered to increase mycobacteriophages’ host range and therapeutic potential against MDR *M. abscessus*.

Recently, compassionate use with phage therapy was included in a pilot study in patients with various mycobacterial infections, untreatable with antibiotics, which showed positive clinical responses in 11 out of 20 patients [135]. Favorable or partial responses were observed in two patients with *M. abscessus* subsp. *massiliense*, six patients with *M. abscessus* subsp. abscessus, one patient with *M. chelonae*, one patient with *Mav* complex, and one patient with disseminated BCG [135]. The development of phage treatments could provide a crucial tool for physicians when no other options are available.

## 5. Novel Therapeutic Strategies

Treatment of *M. abscessus* infection is becoming more challenging with increased resistance to many of the current drugs and the lack of a sufficient pipeline of new drug candidates. 

There are several novel drug approaches, however, that are currently being investigated. Guo et al. have recently shown in vitro efficacy of Cotezolid (MRX-I), (S)-5-([isoxazol-3-ylamino]methyl)-3-(2,3,5-trifluoro-4-[4-oxo-3,4-dihydropyridin-1(2H)-yl]phenyl)oxazolidin-2-one, which is an oxazolidinone, against *M. abscessus* [136]. Linezolid, also an oxazolidinone, is recommended for use against *M. abscessus*; however, Cotezolid may have an advantage by inducing fewer side effects; oral MRX-I administration was found to be well tolerated in humans in a Phase 1 study where adverse events were shown to be mild to moderate [137]. In the study by Guo et al., both Cotezolid (MRX-I) and linezolid are effective against *M. abscessus* but not *M. avium* or *M. intracellulare* [136]. Furthermore, Contezolid (MRX-I) was compatible with other *M. abscessus* drugs, including Azithromycin, Clarithromycin, Cefoxitin, Imipenum, Tigecycline, Bedaquiline, Amikacin, and Amoxifloxacin [136].

An analog of Linezolid, called Sutezolid, exhibits lower in vitro minimal inhibitory concentration (MIC) and minimal bactericidal concentration (MBC) against *M. abscessus* compared to Linezolid (Dae Hun Kim, AAC, 2021, PMID:33903101) and may have fewer in vivo toxicities than linezolid as shown in studies testing the use of these drugs in healthy volunteers for intended use against *Mycobacterium tuberculosis* [138,139]. 

Ganapathy et al. have recently shown that a novel mycobacterial DNA gyrase inhibitor (MGI), an advanced *M. tb* drug candidate, EC/11716, has in vitro bactericidal activity against both Mav and *M. abscessus* and importantly has activity against *M. abscessus* biofilms [140]. EC/11716 was also shown to have in vivo efficacy in a preclinical *M. abscessus* NOD SCID mouse model [140]

Another promising drug candidate for use against *M. abscessus* is T405, which is a novel b-lactam of the penem subclass and was recently shown to have in vitro synergy in combination with other antibiotics, including imipenem cefditoren or avibactam [141]. Furthermore, T405 combined with probenecid exhibited bactericidal efficacy in the C3HeB/FeJ in vivo mouse model against the well-characterized ATCC29977 reference strain (Rimal B. et al., AAC, 2022, PMID:35638855). Beta-lactam antibiotics are known to interfere with bacterial cell wall peptidoglycan biosynthesis (for the mechanism of activity of b-lactams, see a recent review by Turner et al.) [142].

Diazabicyclooctanes (DBOs), including Durlobactam (DUR), are included within a class of novel b-lactamase inhibitors that inhibit peptidoglycan transpeptidases which, when combined with dual b-lactams, could potentially improve clinical efficacy and reduce the toxicity of Mab treatment regimens [143,144]. 

Unlike *M. tuberculosis*, there is currently no vaccine available for *M. abscessus*, and there are no vaccines in clinical development. Therapeutic vaccination as an adjunct to drug treatment against *M. abscessus* and other NTM could shorten drug treatment regimens and decrease the side effects associated with the current repertoire of available drugs used against NTM. 

Recently, two relatively new additions to the anti-NTM drugs, developed originally to treat tuberculosis or leprosy, were used against *M. abscessus*-PD; Bedaquiline (BDQ) and Clofazimine (CFZ), respectively [145,146]. BDQ, an ATPase inhibitor, is the first drug approved to treat MDR-TB by the FDA in 40 years [147]. In a recent study by Sarathy et al., 3,5-dialkoxypyridine analogues of BDQ showed promising in vitro and in vivo activities against *M. abscessus*, similar to its BDQ parent [148]. Given that analogues of BDQ are less lipophilic, have higher clearance, and display lower cardiotoxicity, they are promising drug candidates to be co-administered with currently used drugs. On the other hand, CFZ is an approved drug for leprosy being repurposed for TB treatment [149]. CFZ is considered to be one of few candidates that are being tested for monotherapy against *M. abscessus*-PD [5]. In a recent trial, after one year of CFZ-containing regimes, treatment of *M. abscessus*-PD patients showed conversion to culture negative (CCN) [150]. Both BDQ and CFZ drugs have shown efficacy against *M. abscessus* alone and in combination [145,146]. 

Notably, Amikacin is known to induce systematic toxicity, including hearing loss, loss of balance, or both, especially when given by the intravenous route [151]. To reduce these adverse effects and to increase drug concentrations in endobronchial tissues, amikacin by aerosolization has been increasingly used [152]. In previous trials, inhaled amikacin demonstrated increased efficacy in terms of increased chance of CCN [152,153]. Amikacin liposome inhalation suspension (ALIS), in which amikacin is encapsulated in liposomes and delivered into the lungs via aerosol nebulization, has shown increased efficacy against *M. avium* refractory lung disease [154]. Compared with intravenous administration of non-liposomal amikacin, ALIS increased amikacin concentration by 42-fold in lung tissues, 69-fold in airways and 274-fold in macrophages [155]. ALIS is currently in a phase II trial for treating *M. abscessus*-LD [5]. Recently, a compassionate use study using ALIS in patients with *M. abscessus* pulmonary disease previously treated with various treatment regimens was described. This study included 41 patients, 61% of which had a ‘good outcome’ defined as outcomes cure, microbiologic cure, and clinical cure [156]. 

Another novel approach for treating *M. abscessus*-PD is using the apoptotic bodies to target host immune responses targeted to the pathogen, rather than directly targeting the pathogen. The principal of using apoptotic bodies is to improve phagocytosis, phagolysosomal maturation, and intracellular mycobacterial killing by sending in a second lipid messenger (bioactive lipids) known for promoting phagosomal maturation through recognition of specific lipid-binding domains [157,158]. Apoptotic bioactive-lipids (ABL) loaded with different bioactive lipids have been evaluated in case of bacterial interference of phagolysosome biogenesis and genetically impaired phagolysosome-dependent antimicrobial response, i.e., CF [159]. In both conditions, ABLs demonstrated a significant increase in intraphagosomal acidification and induction of reactive oxygen species (ROS) production and ultimately promoted intracellular mycobacterial killing in macrophages [159]. Recently, in a study by Poerio et al., ABLs loaded with phosphatidylinositol 5-phosphate (ABL/PI5P), alone or in combination with amikacin, have been evaluated for the treatment of *M. abscessus*-PD [160]. The combination treatment of ABL/PI5P and amikacin showed a significant reduction of pulmonary mycobacterial burden. 

BCG, the only approved vaccine used for the prevention of serious forms of TB in children and adolescents, showed cross-protective immunity against *M. avium* and *M. abscessus*-related infection [161]. This fact was confirmed through the epidemiological evidence suggesting that BCG vaccination decreases the risk of developing NTM-PD [162]. It was suggested that BCG vaccination can be used as either a therapeutic or prophylactic vaccine against *M. abscessus*-PD [161,163]. BCG, as a live-attenuated vaccine, induces T-cell expansion important for intracellular pathogens like *M. abscessus* [161,164]. However, BCG is contraindicated for immune-compromised individuals such as HIV/AIDS patients [165,166]. Additionally, BCG does not reduce *M. avium* infection in the mice model in case of prior exposure to NTM [167,168]. The use of BCG as an intervention to prevent or treat *M. abscessus*-PD is hindered by the complex mechanism of NTM exposure. 

Due to the complexities and difficulties in the treatment of *M. abscessus* infection, there is an urgent need for a therapeutic vaccine to overcome the lengthy treatment time and required toxic concentrations of antibiotics [169]. A therapeutic vaccine could also help with the acquired drug resistance to antibiotics used with *M. abscessus* infections. The design of a therapeutic vaccine can be acquired from the knowledge gained in the *M. tb* field. The most prominent example is the use of the Phase 2a clinical trial ID93 + glucopyranosyl lipid adjuvant (GLA) formulated in an oil-in-water stable nanoemulsion (SE) as a therapeutic vaccine against *M. tb* [170,171,172]. This vaccine design highlights the importance of selecting both mycobacterial antigens and a potent immune-stimulating adjuvant. The use of protein/adjuvant immunotherapy combined with a drug treatment strategy is commonly used for other vaccine studies for infectious diseases [170,173,174,175,176,177]. However, the lack of vaccine development for *M. abscessus* infection (either prophylactic or therapeutic) calls for the need to invest in these strategies to overcome the complexities involved with the treatment of *M. abscessus* infections.

## 6. Preclinical Models for *M. abscessus*

The biggest challenge in discovering novel host-directed therapeutic interventions for *M. abscessus* infections is the absence of an adequate experimental animal model. A summary of the preclinical *M. abscessus* models is shown in Table 2. *M. abscessus* are generally less virulent than *Mycobacterium tuberculosis* complex members, shown by the decreased capacity to induce a sustained progressive infection in an immunocompetent mouse model [178]. Therefore, there is an urgent need therefore for the development of a *M. abscessus* challenge model for the development of host-directed therapies and other host interventions such as therapeutic vaccines. Experimental animal models have been of great benefit for developing prophylactic and therapeutic vaccine strategies for treating *M. tb*, which has been the focus of our laboratory for several years [168,169,179,180], and we have now begun applying our expertise to vaccines against NTM infections [168,169]. Others have also focused on this challenging quest for therapeutic solutions against NTM [160,181,182,183,184,185,186]. Many preclinical models have been proposed to study NTM Infection for early drug discovery and vaccine research [30]. 

Regarding *M. abscessus*, many nonmammalian models are also used, such as Amoebas (*Dictyostelium discoideum*) [187], *Drosophila melanogaster* [188], *Galleria mellonella* larvae [189], Silkworm [190], and zebrafish [191,192,193]. Nonmammalian models are valuable models for screening anti-mycobacterial drugs and imaging host–pathogen interactions at a cellular level due to their relative transparency combined with the development of recombinant bacterial strains that express fluorescent proteins. Drawbacks of those models, however, include their inability to mimic chronic infection that can only be modeled in a mammalian host.

The mouse infection model, developed for several infectious disease pathogens, has been more extensively utilized than any other preclinical model for drug discovery and vaccine research. As previously mentioned, the use of an immunocompetent mouse model in *M. abscessus* infection is not considered an adequate model due to the rapid clearance of the *M. abscessus* [194]. The existence of several immunocompromised mouse models, including severe combined immunodeficiency (SCID) mice, granulocyte monocyte-colony stimulating factor knockout mice (GM-CSF−/−), and NOD.CB17-Prkdc^scid^/NCrCrl mice with compromised B cells, T cells, and natural killer cells resulted in *M. abscessus* progressive infection, similar to that seen with human *M. abscessus*-LD [194]. Recently, a protective role for type 1 IFN (IFNβ) has been shown, where *M. abscessus* clearance in macrophages was facilitated through the production of NO in a NO-dependent fashion [195]. The same authors also showed that NOD2-mediated activation of p38 and JNK, ultimately leading to NO production, can effectively clear *M. abscessus* in macrophages. Whereas individual immune factors may be implicated as risk factors for *M. abscessus* infections, the use of transgenic mice with single-gene deletion for NOS, TNF, IFNγ, or MyD88 may be compensated for with a different mechanism of the immune system [195,196,197,198]. Additionally, the route of infection greatly influences the host-immune response against *M. abscessus* lung infection, and so infection results [78,199]. The challenge dose of *M. abscessus* also requires optimization for establishing pulmonary lung infection. An aerosol infection with *M. abscessus* was shown to require 1 × 10^5^–10^9^ CFUs to enable a progressive infection in an immunocompromised mouse model [78].

The use of nude and IFNγ knock-out (GKO) mice present two models in which antibiotic therapy studies can be performed [178]. These animal models, however, are not conducive to studying the efficacy of either prophylactic or therapeutic vaccines against NTM, including *M. abscessus*, for which other mouse models are required. One lesser studied mouse strain for *M. abscessus* infection is the Beige mouse, a model for Chédiak–Higashi syndrome [200], an immune disorder characterized by impaired phagocytosis due to a mutation of a lysosomal trafficking regulator protein [201]. This mouse strain also has defective polymorphonuclear cells, monocytes, and NK cells, with delayed chemotaxis and microbicidal capacity [202]. The Beige model is considered the standard model for much slow-growing NTM, such as *M. avium*, as extreme susceptibility to MAC infection, has been demonstrated by us and others [168,203,204,205]. Previous studies in the Beige mouse model showed a dominant Th2 immunity that allows for MAC growth [206]. Even so, infection of Beige mice with *M. abscessus* does not lead to a persistent infection, which limits its utility as a model for vaccine development [194]. In a previous study on seven mouse strains, including Beige, BALB/c, Nude, GKO, A/J, Swiss, and C57BL/6 mice, most immunocompetent mice were able to rapidly clear the infection with *M. abscessus* (by 30 days in the lungs and 60 days in the spleen) [178]. Notably, those infections were done via the intravenous route, decreasing the chances for progressive infection. Additionally, the laboratory *M. abscessus* ATCC 19977 strain is often utilized in published studies. Future considerations could aim to establish progressive infection through using the aerosol route and through the use of a clinical isolate, which would likely show a higher degree of virulence. Furthermore, one could compare infection with the S and R variants of *M. abscessus* in the Beige mice to further shed light on the mechanism(s) by which the R variant is more virulent than the S variant in an in vivo model.

## Figures and Tables

**Figure 1 microorganisms-10-01454-f001:**
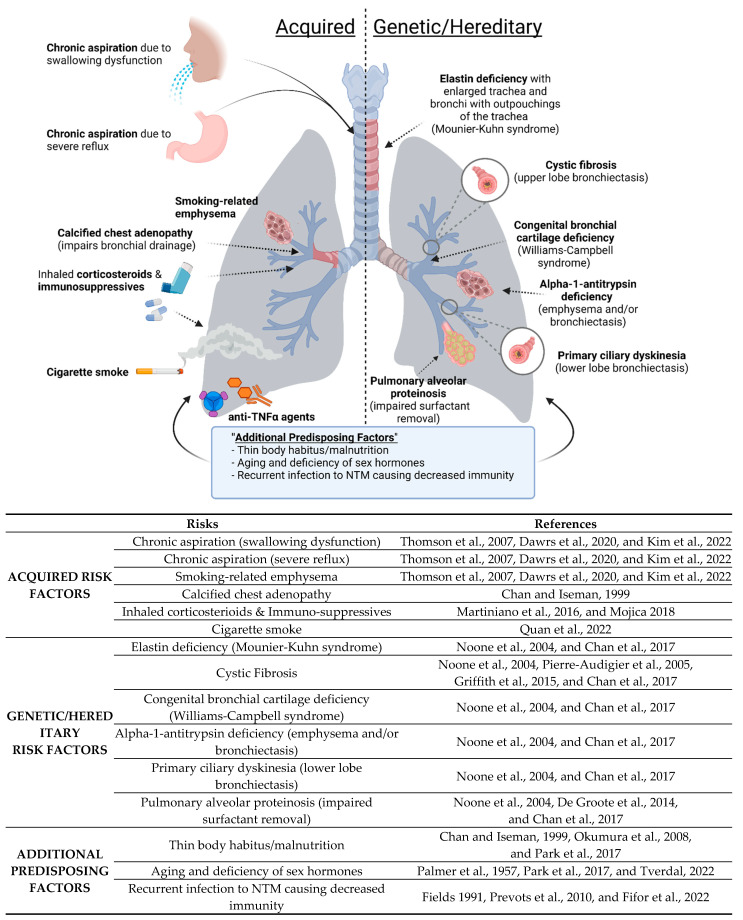
Risk factors for NTM lung disease [6,16,31,32,33,35,36,38,40,41,42,43,44,45,46,47,48,49,50,51].

**Figure 2 microorganisms-10-01454-f002:**
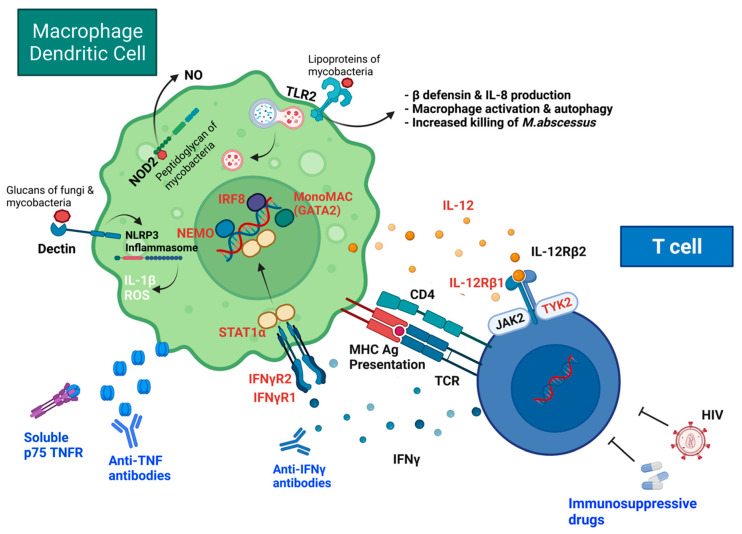
Genetic and acquired causes of disseminated NTM disease.

**Figure 3 microorganisms-10-01454-f003:**
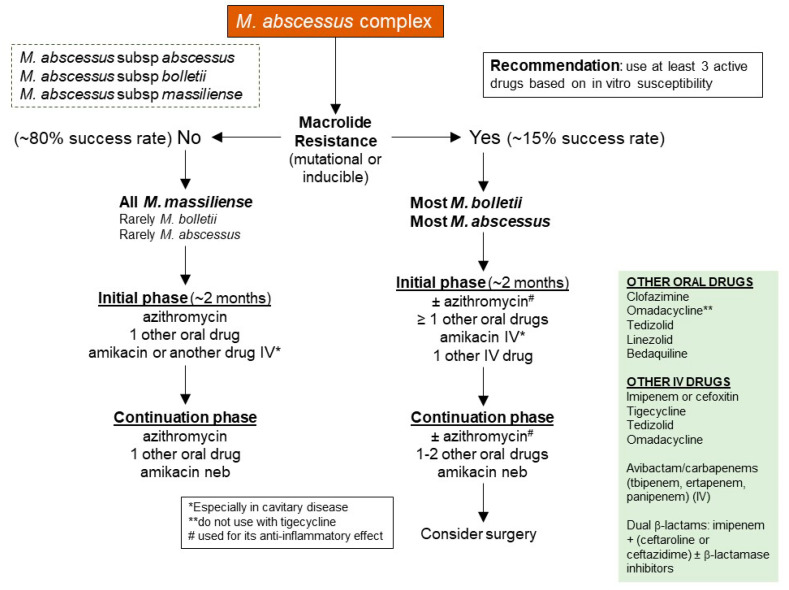
A schematic representation of *M. abscessus* current treatment protocol.

**Figure 4 microorganisms-10-01454-f004:**
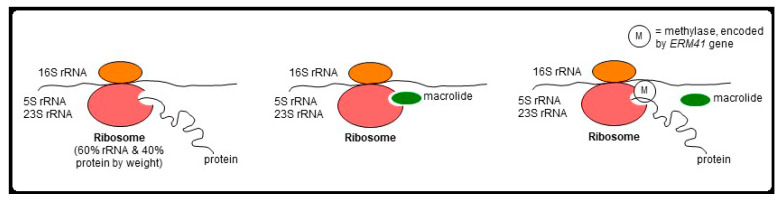
Mechanism of inducible resistance in *M. abscessus*. In *M. abscessus* sensu stricto and *M. bolletii*, macrolide binds to 23S rRNA and inhibits bacterial protein synthesis. With induction of methylase production by clarithromycin, the methylase prevents the binding of macrolide, creating an inducible resistance. Since clarithromycin induction of the ERM41 gene to produce methylase is much greater than by azithromycin, clarithromycin is much more likely to induce macrolide resistance than azithromycin.

**Table 1 microorganisms-10-01454-t001:** Major known mechanisms that predispose to NTM-LD.

Mechanism for Predisposition	Associated Conditions
**Structural lung disease**	Emphysema, bronchiectasis of any cause, including CF, alpha-1-antitrypsin deficiency (AAT), and Sjogren’s syndrome
**Primary or ciliary dysfunction**	Primary ciliary dyskinesia, bronchiectasis of any cause, MST1R dysfunction (?)
**Thickened secretions**	CF
**Macrophage dysfunction**	Alpha-1-antitrypsin (AAT) deficiency or anomaly, silica exposure, pulmonary alveolar proteinosis
**Deficiency of specific immune molecules**	Anti-TNF therapy, common variable immunodeficiency, CF (human beta-defensin?), inhaled glucocorticoid
**Cartilage deficiency in airways**	Williams–Campbell syndrome
**Elastin deficiency in airways**	Mounier-Kuhn syndrome

MST1R = macrophage-stimulating 1 receptor.

**Table 2 microorganisms-10-01454-t002:** Summary of different models used for *M. abscessus* infection.

Type of Model	Model	Nature	Advantages	Drawbacks
**Nonmammalian models**	Amoebas (*Dictyostelium discoideum*)	- Environmental phagocyte organisms - Natural hosts of NTM organisms	- Model for host-pathogen interaction - Used for screening anti-mycobacterial drugs - Relative transparency	- Maximal survival temperature is 27 °C which may affect bacterial growth - Inability to mimic chronic infection
	*Drosophila melanogaster*	- Adult ages 5 to 7 days are used as models for *M. abscessus* infection	- Host for *M. abscessus* infection - Used for screening anti-mycobacterial drugs	- Minimal pathogenicity after *M. abscessus* infection
	*Galleria mellonella* larvae	- Larvae are used as models for studying the innate immune system	- Physiologic temperatures (up to 37 °C) suitable for bacterial growth - Relative transparent Used for screening anti-mycobacterial drugs	- Drug-exposure response doesn’t emulate mammalian host Inability to mimic chronic infection
	Zebrafish	- Model for early innate immunity given by macrophages and neutrophils - Mycobacteria-infected zebrafish mimics granuloma-like lesions	- Used for screening anti-mycobacterial drugs - Relative transparent	- Susceptibility profiles to different mycobacterial organisms are different
	Silk worm	- Larvae are used as models for studying bacterial infections	- Used for screening anti-mycobacterial drugs	- Rapid growing NTM are detrimental for larvae
**Mammalian models**	Nude Mice	- Compromised B cells, T cells and natural killer cells	- Similar progressive infection with human *M. abscessus* lung disease	- Can’t be used for studying the efficacy of either prophylactic or therapeutic vaccines
	GKO Mice	- Ifnγ knock out	- Similar progressive infection with human *M. abscessus* lung disease	- Can’t be used for studying the efficacy of either prophylactic or therapeutic vaccines
	Beige Mice	- Mutation of a lysosomal trafficking regulator protein leading to impaired phagocytosis	- Extreme susceptibility to MAC - Can be used for studying the efficacy of vaccines	- Less studied mouse model for *M. abscessus* infection
	C57BL/6 Mice	- Susceptible to NTM infection	- Can be used for studying the efficacy of vaccines	- Rapid clearance of the *M. abscessus*
	BALB/c	- Susceptible to NTM infection	- Can be used for studying the efficacy of vaccines	- Rapid clearance of the *M. abscessus*

## Data Availability

Not applicable.

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
