# Peer review of "Mycobacterium abscessus: It’s Complex"

_microorganisms, 2022, doi:10.3390/microorganisms10071454_

Round 1

Reviewer 1 Report

The review is very interesting; however, the following points should be raised before publication:

-        Abstract: Name of the bacteria is wrong, the authors should replace “abscesses” by “abscessus”

-        The new nomenclature that has been endorsed by the International Committee on Systematics of Prokaryotes should be mentioned and discussed (see International code of nomenclature of prokaryotes. Int J Syst Evol Microbiol 2019; 69: S1–S111.) in the light of Tortoli’s publication (Tortoli E, Brown-Elliott BA, Chalmers JD, et al. Same meat, different gravy: ignore the new names of mycobacteria. Eur Respir J 2019; 54: 1900795 [https://doi.org/10.1183/13993003.00795-2019].

-        Line 46: Table 1 does not list MAC subspecies

-        Figure 1:

o   the authors should add references into the risk factirs mentioned in the figure

o   not all the risks factors mentioned in the figure are explained in the texte

-        Chapter “treatment against Mabs related infection”: the differences in natural susceptibility to clarithromycin between Mabs subspecies should be added in this chapter and the last European recommendations for NTM treatment cited (Lange C, Böttger EC, Cambau E, Griffith DE, Guglielmetti L, van Ingen J, Knight SL, Marras TK, Olivier KN, Santin M, Stout JE, Tortoli E, Wagner D, Winthrop K, Daley CL; expert panel group for management recommendations in non-tuberculous mycobacterial pulmonary diseases. Consensus management recommendations for less common non-tuberculous mycobacterial pulmonary diseases. Lancet Infect Dis. 2022 Jul;22(7):e178-e190. doi: 10.1016/S1473-3099(21)00586-7. Epub 2022 Jan 25. Erratum in: Lancet Infect Dis. 2022 Mar;22(3):e73. PMID: 35090639.)

-        Chapter “novel strategies for treating drug-resistant Mabs”: the authors should also mention the issue due to resistance to clarithromycine, which is also a key drug for the treatment of infections due to Mabs.

-        Chapter “preclinical models for Mabs” is very dense. It would be easier to read if associated to a table summarizing what is known, the drawbacks, advantages and achievements of the different models that have been used.

-        Chapter “novel therapeutic strategies ”does not mention some recent work  that could be interesting such as contezolid or beta-lactams (Ganapathy et al, AAC, 2021, Guo et al, AAC, 2021; Rimai et al, AAC, 2022, Nguyen, CID, 2021 …). It is also somehow redundant with the chapter “novel strategies for treating drug-resistant Mabs”, both could be merged in a single chapter.

Author Response

Reviewer 1

The review is very interesting; however, the following points should be raised before publication:

  1. Abstract: Name of the bacteria is wrong; the authors should replace “abscesses” by “abscessus”

Response: The name was corrected to Mycobacterium abscessus throughout the manuscript.

  1. The new nomenclature that has been endorsed by the International Committee on Systematics of Prokaryotes should be mentioned and discussed (see international code of nomenclature of prokaryotes. Int J Syst Evol Microbiol 2019; 69: S1–S111.) in the light of Tortoli’s publication (Tortoli E, Brown-Elliott BA, Chalmers JD, et al. Same meat, different gravy: ignore the new names of mycobacteria. Eur Respir J 2019; 54: 1900795 [https://doi.org/10.1183/13993003.00795-2019].

Response: We agree with the reviewer and have selected to choose the nomenclature that is less prone to errors, and one that is based on the single genus, ‘Mycobacterium’ for the descriptions of NTM throughout our manuscript. We agree that renaming clinically important nontuberculous mycobacteria would be confusing to both researchers and clinicians. [We have added the reference, Tortoli E. et al., Eur Resp J., 2019, 54 to the first instance we use ‘Mycobacterium’].

  1. Line 46: Table 1 does not list MAC subspecies

Response: The text inadvertently referencing Table 1 was removed.

  1. Figure 1: 
    1. the authors should add references into the risk factors mentioned in the figure
    2. not all the risks factors mentioned in the figure are explained in the text

Responses: We have added references to a Table under Figure 1 along with the risk factors and have included extra text describing the risk factors mentioned in Figure 1.

  1. Chapter “treatment against Mabs related infection”: the differences in natural susceptibility to clarithromycin between Mabs subspecies should be added in this chapter and the last European recommendations for NTM treatment cited (Lange C, Böttger EC, Cambau E, Griffith DE, Guglielmetti L, van Ingen J, Knight SL, Marras TK, Olivier KN, Santin M, Stout JE, Tortoli E, Wagner D, Winthrop K, Daley CL; expert panel group for management recommendations in non-tuberculous mycobacterial pulmonary diseases. Consensus management recommendations for less common non-tuberculous mycobacterial pulmonary diseases. Lancet Infect Dis. 2022 Jul;22(7):e178-e190. doi: 10.1016/S1473-3099(21)00586-7. Epub 2022 Jan 25. Erratum in: Lancet Infect Dis. 2022 Mar;22(3):e73. PMID: 35090639.)

Response: References were added to line 559. Figure 4 and the second paragraph describe the differences associated with clarithromycin between the Mabs subspecies.

  1. Chapter “novel strategies for treating drug-resistant Mabs”: the authors should also mention the issue due to resistance to clarithromycin, which is also a key drug for the treatment of infections due to Mabs.

Response: We agree.  The following text has been added to Section II (Strategies for treating drug resistant Mabs) under “Treatment against Mabs related infection”:  “Currently, the macrolides (clarithromycin or azithromycin) are the most used antibiotics against Mabs.  Thus, it is not surprising that whether an Mabs isolate is susceptible or resistant to the macrolides is a key decision point in both the initial choice of antibiotic regimen and clinical outcome (Figure 3).  While absence or presence of macrolide resistance is also a key decision point for treatment and outcome of MAC-LD, the option for oral antibiotics for Mabs is much more limited than for MAC.”

  1. Chapter “preclinical models for Mabs” is very dense. It would be easier to read if associated to a table summarizing what is known, the drawbacks, advantages and achievements of the different models that have been used.

Response: We agree with this statement. Table 2 was added summarizing different models

  1. Chapter “novel therapeutic strategies ”does not mention some recent work  that could be interesting such as contezolid or beta-lactams (Ganapathy et al, AAC, 2021, Guo et al, AAC, 2021; Rimai et al, AAC, 2022, Nguyen, CID, 2021 …). It is also somehow redundant with the chapter “novel strategies for treating drug-resistant Mabs”, both could be merged in a single chapter.

Response: We agree with this assessment and have included novel drug treatment strategies along with the suggested references.

Reviewer 2 Report

This manuscript focused on Mycobacterium abscessus (Maps), which appeared a very intriguing and important topic for readers. Authors well summarized many issues for combating this NTM. Therefore, this manuscript will be worthy acceptable as a review in this journal. But following points will be considered in the revised manuscript.

Major points:

1) Authors should focus on Mabs in this review. But I think authors described characteristics common to NTM (not only Mabs but also MAC) in most sections even though they described about Mabs. Therefore, authors should add explanation about difference or similarity between Mabs and other NTM (particularly MAC) in such sections. This will make this review much sharper. (For example, Figure 3 is the case for Mabs treatment. Which part is different from MAC treatment?)

2) Regarding Figure 2, more detail explanation is needed in the text.

3) Figure 4 is not correct. Is methylase bound to macrolide binding site? Figure 4 is not corresponding to the explanation in line 333-334. No rrl in the figure. What is rrl?

4) I cannot understand the part of leptin (line 222-229). Fat mice (leptin deficiency) are more susceptible to Mabs. Young women with anorexia nervosa are more susceptible to Mabs.

Minor points:

5) The spell "M. abscesses" in the title and keywords is OK?

6) Line 341;Mabs-PD, what abbreviation is PD (cannot find it)? rrs mutant, what is rrs?

7) Line 416-422; silkworm infection assays have been reported with NTM such as M. abscessus (Hosoda K. et al. Molecules 25, 4972 (2020)) and M. avium (Yagi et al. Drug Discoveries Therapeutics 14, 287-295 (2021)). The articles should be cited and added to References.

8) Line 502-513 about Amikacin liposome (ALIS); The information of ALIS seemed old. It was approved by FDA (US) on Oct, 2018, and also in Japan on March, 2021. Now ALIS is clinically used. So more updated information is needed.

9) Five citations (Jeon et al., Lavoilay et al., Kathavade et al., Candido et al. and Cowman et al.) appeared in line 76-78 cannot be found in  References. I cannot follow the point of Ref. 85 (Kim, Song Yee et al.) in the text. Check these references.

Author Response

This manuscript focused on Mycobacterium abscessus (Mabs), which appeared a very intriguing and important topic for readers. Authors well summarized many issues for combating this NTM. Therefore, this manuscript will be worthy acceptable as a review in this journal. But following points will be considered in the revised manuscript.

Major points:

  • Authors should focus on Mabs in this review. But I think authors described characteristics common to NTM (not only Mabs but also MAC) in most sections even though they described about Mabs. Therefore, authors should add explanation about difference or similarity between Mabs and other NTM (particularly MAC) in such sections. This will make this review much sharper. (For example, Figure 3 is the case for Mabs treatment. Which part is different from MAC treatment?)

Response: With regards to Figure 3 of Mabs treatment and how it differs from MAC treatment. The following text has been added to Section II (Strategies for treating drug resistant Mabs) under “Treatment against Mabs related infection”:  “Currently, the macrolides (clarithromycin or azithromycin) are the most used antibiotics against Mabs.  Thus, it is not surprising that whether an Mabs isolate is susceptible or resistant to the macrolides is a key decision point in both the initial choice of antibiotic regimen and clinical outcome (Figure 3).  While absence or presence of macrolide resistance is also a key decision point for treatment and outcome of MAC-LD, the option for oral antibiotics for Mabs is much more limited than for MAC.”

  • Regarding Figure 2, more detail explanation is needed in the text.

Response: More extensive text related to Figure 2 is included within the text.

  • Figure 4 is not correct. Is methylase bound to macrolide binding site? Figure 4 is not corresponding to the explanation in line 333-334. No rrl in the figure. What is rrl?

Response: We apologize for lack of clarity.  There are two types of potential macrolide resistance with Mabs, “genetic resistance” and “inducible resistance.”  Figure 4 is a diagram showing only the mechanism of inducible macrolide resistance.  We have clarified the text describing the two forms of macrolide resistance in Mabs species under “Treatment against Mabs related infections (Section I. Antibiotics used for treating Mabs)”.

“Macrolides target bacterial 23S rRNA, inhibiting bacterial protein synthesis.  Mabs possess two major forms of macrolide resistance and both involve the bacterial 23S rRNA by different mechanisms.  The first is “genetic” macrolide resistance and is due to single point mutation in position 2058 or 2059 of the bacterial 23S rRNA gene (also known as the rrl gene) (Carneiro et al., 2017).  The second form is known as “inducible” macrolide resistance wherein a functional ERM41 gene encodes a methylase that occupies a site on 23S rRNA preventing macrolides from binding (Choi et al., 2012; Nash et al., 2005) (Figure 4).  Among the Mabs organisms, the majority of subsp abscessus and subsp bolletii strains possess a functional ERM41 gene, which confers an inducible resistance to macrolides.

“Conversely, a minority (15-20%) of subsp abscessus isolates possess a T28C mutation of the ERM41 gene, resulting in a non-functional methylase with preserved macrolide susceptibility.  Similarly, all subsp massiliensestrains contain a partially deleted, non-functional ERM41 gene and thus also have preserved macrolide susceptibility.  Thus, in the absence of rrl gene mutation, NTM-LD patients infected with Mabs with a non-functional ERM41 gene and hence preserved macrolide susceptibility (a minority of subsp abscessus strains and all strains of subsp massiliense) have better clinical outcome than those infected with Mabs isolates with a functional ERM41 gene and consequently inducible macrolide resistance (most subsp abscessus and essentially all of subsp bolletii).”

  • I cannot understand the part of leptin (line 222-229). Fat mice (leptin deficiency) are more susceptible to Mabs. Young women with anorexia nervosa are more susceptible to Mabs.

Response: We cannot find the line saying ‘fat mice (leptin deficiency) are more susceptible to Mabs. The opposite is true. Slender individuals have abnormal expression of leptin and adiponectin. Deficiency of leptin leads to NTM susceptibility. We have stated that ‘Indeed, mice deficient in leptin are more susceptible to experimental Mabs lung infection (Lord et al., 1998; Diane Ordway et al., 2008)’. We have also included another reference to help with this concept (Kartalija M. et al, 2012, PMID:23144328).

  1. Minor points:
  • The spell "M. abscesses" in the title and keywords is OK?

Response: The spelling has been corrected; this should be M. abscessus.

  • Line 341; Mabs-PD, what abbreviation is PD (cannot find it)? rrs mutant, what is rrs?

Response: PD stands for ‘pulmonary disease’. This is now defined in the text

  • Line 416-422; silkworm infection assays have been reported with NTM such as M. abscessus (Hosoda K. et al. Molecules 25, 4972 (2020)) and M. avium (Yagi et al. Drug Discoveries Therapeutics 14, 287-295 (2021)). The articles should be cited and added to References.

Response: The silkworm model was added (line 575) and Table 2.

  • Line 502-513 about Amikacin liposome (ALIS); The information of ALIS seemed old. It was approved by FDA (US) on Oct, 2018, and also in Japan on March, 2021. Now ALIS is clinically used. So more updated information is needed.

Response: We agree with this assessment, the following text was added at line 481-484

“Recently, a compassionate use study using ALIS in patients with Mabs pulmonary disease previously treated with various treatment regimens was described. This study included 41 patients, 61% of which had a ‘good outcome’ defined as outcomes cure, microbiologic cure, and clinical cure. PMID: 35063454”

  • Five citations (Jeon et al., Lavoilay et al., Kathavade et al., Candido et al. and Cowman et al.) appeared in line 76-78 cannot be found in References. I cannot follow the point of Ref. 85 (Kim, Song Yee et al.) in the text. Check these references.

Response: All citations were revised, and missing references were added

Reviewer 3 Report

The manuscript presented to me for review, entitled "Mycobacterium abscesses: it's complex." by Hazem F. M. Abdelaal et al. very meticulously and clearly describes the problem of Mycobacterium abscesses (Mabs) as an opportunistic pathogen.

The authors, however, did not avoid some shortcomings in their work:

·      Abstract different types of fonts necessary to standardize

·      Citation style inconsistent with journal guidelines

·      The resolution of the figures needs to be improved

·      Table 1 bold header

·      Standardizing the headings of chapters and subsections and their numbering method

·      Line 260 and 271 repeated citation

·      Line 279, a different citation style

·      Line 404 redundant characters

·      Line 405 has a different type of font. Authors should revise the entire manuscript and standardize the font following the journal's guidelines.

·      Necessary to adapt the bibliography to the journal's requirements.

·      Authors should update the cited articles whenever possible as they repeatedly cite works published before the year 2000.

Author Response

The manuscript presented to me for review, entitled "Mycobacterium abscesses: it's complex." by Hazem F. M. Abdelaal et al. very meticulously and clearly describes the problem of Mycobacterium abscesses (Mabs) as an opportunistic pathogen.

The authors, however, did not avoid some shortcomings in their work:

  • Abstract different types of fonts necessary to standardize

Response: Abstract and all manuscript font was checked, and proper correction was taken.

  • Citation style inconsistent with journal guidelines

Response: Citation was fixed following the Instructions for Authors of microorganism journal

  • The resolution of the figures needs to be improved

Response: Resolution of Figures 1 and 2 have been improved.

  • Table 1 bold header

Response: The font on the Table header were changed to Bold.

  • Standardizing the headings of chapters and subsections and their numbering method

Response: All headings and subheadings were numbered.

  • Line 260 and 271 repeated citations

Response: These repeated citations have been corrected.

  • Line 279, a different citation style

Response: The citation style has been corrected.

  • Line 404 redundant characters

Response: The redundant characters were removed.

  • Line 405 has a different type of font. Authors should revise the entire manuscript and standardize the font following the journal's guidelines.

Response: The font was adjusted as mentioned above.

  • Necessary to adapt the bibliography to the journal's requirements.

Response: Citation was fixed following the Instructions for Authors of microorganism journal

  • Authors should update the cited articles whenever possible as they repeatedly cite works published before the year 2000.

Response: Proper citations were updated according to each point.

Reviewer 4 Report

With this review the authors aim to outline the critical issues and challenges in the treatments of Mab infections. The topic is highly relevant due to the increasing issue of NTM infections, particularly of Mab. After a brief overview of the virulence factors and pathogenicity of Mab, they describe the different typology of NTM infection, with a particular focus on the host conditions that favour such infections. Finally, the possible new treatment strategies and the advancements in preclinical models are reported.

Overall, the review appears quite unbalanced, focusing more on the hosts and on its predisposing conditions to infection, rather then on the pathogen or on the host-pathogen interaction. Considering the aim and scope of the journal, it would take a more detailed discussion about virulence factors, biofilm, mechanisms of resistance as well as their consequence on the interaction with the host.

The manuscript should be better organized and with more accuracy:

-   In title, keywords and abstract, It would be better to use Mycobacterium abscessus rather than Mycobacterium abscesses. Moreover, in the title it should be in italic font.

-   There is a table missing: in line 45-46 the authors state: “The opportunistic slow 44 growers include the M. avium complex (MAC), with several different species including M. avium, M. intracellulare subsp intracellulare and subsp chimaera, among others listed in Table 1.”  But table 1 reports the Major known mechanisms that predispose to NTM-LD . Anyway, table 1 is not in the correct position.

-   The legend of figure 2 should be more detailed and the figure better explained. For instance this figure presents several abbreviations that are not defined

-   The references are not reported in a homogeneous form, in some cases as authors and years i.e.: (Johansen et al., 2020) in other as number (30), or in an undefined mode (Fibrosis, 1999).

-   In the reference list some are incomplete (missing volume, pages or article number……..) i.e. 1,2, 5, 27, 98……. Some references are incorrectly reported (i.e. 55, 112) Moreover some references are not pertinent, ie. ref 112. Moreover, the style is completely not homogeneous.

Author Response

With this review the authors aim to outline the critical issues and challenges in the treatments of Mab infections. The topic is highly relevant due to the increasing issue of NTM infections, particularly of Mab. After a brief overview of the virulence factors and pathogenicity of Mab, they describe the different typology of NTM infection, with a particular focus on the host conditions that favor such infections. Finally, the possible new treatment strategies and the advancements in preclinical models are reported.

Overall, the review appears quite unbalanced, focusing more on the hosts and on its predisposing conditions to infection, rather than on the pathogen or on the host-pathogen interaction. Considering the aim and scope of the journal, it would take a more detailed discussion about virulence factors, biofilm, mechanisms of resistance as well as their consequence on the interaction with the host.

The manuscript should be better organized and with more accuracy:

  1. In title, keywords and abstract, It would be better to use Mycobacterium abscessus rather than Mycobacterium abscesses

Response: This has been corrected.

  1. There is a table missing: in line 45-46 the authors state: “The opportunistic slow 44 growers include the M. avium complex (MAC), with several different species including M. avium, M. intracellulare subsp intracellulare and subsp chimaera, among others listed in Table 1.” But table 1 reports the Major known mechanisms that predispose to NTM-LD. Anyway, table 1 is not in the correct position.

Response: The reference to Table 1 was inadvertently included and has been removed.

  1. The legend of figure 2 should be more detailed and the figure better explained. For instance, this figure presents several abbreviations that are not defined.

Response: More extensive text related to Figure 2 is included within the text.

  1. The references are not reported in a homogeneous form, in some cases as authors and years i.e.: (Johansen et al., 2020) in other as number (30), or in an undefined mode (Fibrosis, 1999).

Response: All references have been corrected and are homogenous.   

  1. In the reference list some are incomplete (missing volume, pages or article number……..) i.e. 1,2, 5, 27, 98……. Some references are incorrectly reported (i.e. 55, 112) Moreover some references are not pertinent, ie. ref 112. Moreover, the style is completely not homogeneous.

Response: All references have now been corrected and the style is consistent and homogenous.  

Round 2

Reviewer 1 Report

The authors did not answer and/or correct their manuscript as requested previously. For example, they added only a reference without any explanation regarding the nomenclature changes, which  is not enough.

Reviewer 4 Report

The revised form of the manuscript is suitable for pubblication.